# Characteristics of Nontyphoid *Salmonella* Isolated from Human, Environmental, Animal, and Food Samples in Burkina Faso: A Systematic Review and Meta-Analysis

**DOI:** 10.3390/antibiotics13060556

**Published:** 2024-06-13

**Authors:** Kuan Abdoulaye Traore, Abdoul Rachid Aboubacar-Paraiso, Soutongnooma Caroline Bouda, Jean Bienvenue Ouoba, Assèta Kagambèga, Pierre Roques, Nicolas Barro

**Affiliations:** 1Laboratoire de Biologie Moléculaire, d’Epidémiologie et de Surveillance des Bactéries et Virus Transmissibles par les Aliments (LaBESTA), Université Joseph KI-ZERBO (UJKZ), Ouagadougou 03 BP 7021, Burkina Faso; rachidparaiso56@gmail.com (A.R.A.-P.);; 2Laboratoire Sciences de la Vie et de la Terre (LaSVT), Université Norbert ZONGO (UNZ), Koudougou BP 376, Burkina Faso; 3Centre Universitaire de Manga (CUM), Université Norbert ZONGO (UNZ), Koudougou BP 376, Burkina Faso; 4Department of Biology, Institute of Sciences (IDS), Ouagadougou 1757, Burkina Faso; 5Virology Unit, Institut Pasteur de Guinée (IPGui), Conakry 4416, Guinea; pierre.roques@pasteur.fr

**Keywords:** *Salmonella* spp., nontyphoidal, serotypes, resistance phenotype, systematic review, meta-analysis, Burkina Faso

## Abstract

*Salmonella* is one of the world’s leading causes of zoonotic and foodborne illnesses. Recently, antimicrobial resistance (AMR) has become one of the most critical challenges to public health and food safety. Herein, we employed a meta-analysis to determine the pooled prevalence and spatiotemporal distribution of serovars and antimicrobial resistance in NTS in Burkina Faso. To find eligible articles, a comprehensive literature search of PubMed, African Journals Online, ScienceDirect, Google Scholar, and the gray literature (university libraries) in Burkina was conducted for the period from 2008 to 2020. Studies meeting the inclusion criteria were selected and assessed for risk of bias. To assess the temporal and spatial relationships between serotypes and resistant strains from humans, animals, food, and the environment, a random-effects statistical model meta-analysis was carried out using the Comprehensive Meta-Analysis Version 3.0 program. The NTS prevalence rates were 4.6% (95% CI: 3–7) and 20.1% (95% CI: 6.6–47.4) in humans and animals, respectively, and 16.8% (95% CI: 10.5–25.8) and 15.6% (95% CI: 8.2–27.5) in food and the environment, respectively. Most NTS serovars were *S.* Derby, reported both in food and animals, and *S.* Typhimurium, reported in humans, while *S.* Croft II, *S.* Jodpur II, and *S.* Kentucky were the most prevalent in the environment. NTS isolates were highly resistant to erythromycin, amoxicillin, cefixime, and cephalothin, with a pooled prevalence of multidrug resistance of 29% (95% CI: 14.5–49.5). The results of this review show a high diversity of *Salmonella* serotypes, as well as high antibiotic resistance in *Salmonella* isolates from animal, human, food, and environmental samples in Burkina, calling for a consolidated “One Health” approach to better understand the drivers of pathogen emergence, spread, and antimicrobial resistance, as well as the formulation of intervention measures needed to limit the risk associated with the disease.

## 1. Introduction

Urbanization and the impact of socioeconomic development have become the main determinants of the health of the world’s population and the hygiene of the environment in which they live [1]. A lack of hygiene thus provides bioecological conditions favorable for the development of pathogenic germs responsible for numerous diseases, particularly gastroenteritis, the main manifestation of which is diarrhea [2]. There are multiple causes of gastroenteritis, including bacteria, viruses, and parasites [3]. Among bacterial etiological agents, *Salmonella* spp. is among the most problematic food-borne and zoonotic pathogens threatening general health and well-being [4].

*Salmonella* is the main cause of acute gastroenteritis in many countries, and salmonellosis remains a major public health problem worldwide, particularly in developing countries [5]. It manifests primarily as mild diarrhea, also known as food poisoning [6].

The global burden of nontyphoidal *Salmonella* gastroenteritis is estimated to be 93.8 million cases of gastroenteritis each year, with 155,000 deaths [7]. Moreover, it remains the second most reported agent of gastrointestinal infections in humans in Europe and North America [8].

In Africa, it has consistently been reported as a leading cause of bacteremia in immunocompromised individuals, infants, and newborns [9]. Nontyphoidal *Salmonella* serotypes (NTS) are among the most frequent causes of bloodstream infections in sub-Saharan Africa, with approximately 680,000 deaths per year, mainly in children under five years of age [10]. Although more than 2700 serotypes of *Salmonella enterica* have been identified [11], the serotypes most commonly implicated in invasive disease are *S.* Typhimurium, *S.* Enteritidis, and *S.* Dublin [12]. Nevertheless, in recent years, data have revealed the emergence of infection due to the consumption of raw fruits and vegetables associated with rare serotypes from the environment [13].

Prevention of salmonellosis (caused by nontyphoidal *Salmonella*) is difficult due to its complex epidemiology and multiple modes of transmission [14]. Human disease can result from exposure to numerous sources, such as infected animals, contaminated foodstuffs, contaminated water, and direct contact with an infected environment or direct contact between humans [15].

Poultry and pork meats, eggs, dairy products, and green vegetables contaminated with manure or irrigation water are the agents or risk factors most cited in the transmission of this bacterium [9].

The public health problem associated with nontyphoidal *Salmonella* is antimicrobial resistance. The emergence of multiple antibiotic-resistant *Salmonella* strains adds another important dimension to the challenge of controlling *Salmonella*, as resistant variants can compromise the ability to treat human infections, a particularly important issue in the case of systemic infections [14]. Several clones of multidrug-resistant *Salmonella* emerged in the late 1990s and early 2000s, and since then, their prevalence in humans, domestic animals, and other wildlife has spread globally [16,17,18]. Recently, the increasing prevalence of *Salmonella* multi-resistant to clinically important antimicrobials, such as fluoroquinolones and third-generation cephalosporins, has become an emerging problem worldwide [19,20,21,22].

Despite the knowledge of the prevalence of *Salmonella* and its antimicrobial resistance profile, which is mostly reported by individual and local surveillance study(s), comprehensive and robust studies of the prevalence and antimicrobial resistance pattern in Burkina are poorly characterized [23,24,25,26]. Therefore, concise information from a systematic review is more helpful for scientific users to identify gaps for additional studies and for policymakers to develop prevention and control strategies based on the scientific information provided. Thus, this meta-analysis includes a comprehensive evaluation of the scientific literature published between 2008 to 2022 on the state of knowledge on the prevalence, serovars, and antimicrobial resistance phenotypes of NTS strains from studies carried out in Burkina Faso.

## 2. Results

### 2.1. Overview of the Selected Studies

A total of 85 articles were found in our initial research, of which 70 publications were from Google Scholar, PubMed, Science Direct, and Research Gate, and 15 were from other sources. Of the total, 28 publications (33%) considered suitable for inclusion in this review were identified, including 23 eligible studies from a database search (85.71%) and 5 from the literature searches at university libraries (17.86%). A flow diagram of the literature search and selection of eligible studies is presented in Figure 1.

### 2.2. Study Characteristics of Eligible Studies

The following are the characteristics of the studies that are eligible: articles published primarily on the quantitative prevalence of *Salmonella* serovars/isolates in humans, animals (chickens, ducks, cows, pigs, sheep, etc.), foods (lettuce, sandwiches, beef, mutton, poultry, etc.), and the environment in Burkina Faso; the type of samples and method of diagnostics used; the exact number of samples, as well as the number of positives tested; articles reported in English only; and antibiotic resistance. All of the journal articles were published between 2008 and 2022 (Table 1).

All of the studies included in this review were carried out in towns located in four regions on the country, Centre, Nord, Hauts-Bassins and Boucle du Mouhoun. Several studies were from Ouagadougou (*n* = 24), followed by Area 1 (Gourcy and Boromo (*n* = 3)); Area 2 (Bobo-Dioulasso and Ouagadougou (*n* = 2)); Area 3 (Ouagadougou, Gourcy and Boromo (*n* = 1)); Nanoro (*n* = 1); and Nouna in the province of Kossi (*n* = 1) (Figure 2). The most common method for determining the AR profiles of *Salmonella* serotypes isolated from all of the studies included in this systematic and meta-analysis study was disk diffusion.

Of the twenty-eight included studies, twelve (12) were from food sources, eleven (11) were from clinical sources (mainly children under 5 years of age), two (02) were from animal sources, one (01) was from environmental sources, and one (01) included both animal and environmental sources (Figure 3). One study examined food, human, and environmental hosts, and one study included hosts of both food and animal origin (Figure 1).

### 2.3. Prevalence Based on Sources and Locality (Sampling Sites)

A high degree of heterogeneity was observed in studies carried out on different sources (Table 2). For the animals, pooled heterogeneity was observed [20.1% (95% CI: 6.6–47.4), I^2^ = 98.48]; for food [16.6% (95% CI: 9.9–26.4), I^2^ = 96.10]; for humans [5.4% (95% CI: 4.8–6), I^2^ = 91.94]; and for the environment [15.6% (8.2–27.5), I^2^ = 83.20]. Egger’s test revealed no statistically significant publication bias in the estimation of the prevalence of *Salmonella* from three sources: humans (*p* = 0.17), food (*p* = 0.14), and the environment (*p* = 0.45). However, one source of heterogeneity was identified as significant (*p* = 0.03). The pooled prevalence of *Salmonella* isolates found in Ouagadougou was 23.1% (95% CI: 21.9–24.4, I^2^ = 98.25], followed by area 2 [18.4% (95% CI: 14.6–22.8), I^2^ = 0.00] and area 1 [6.4% (95% CI: 4.9–8.2), I^2^ = 70.92]. However, some study areas and cities (areas 3, Nanoro, and Nouna) were not included in the meta-analyses due to the low number of studies.

### 2.4. Pooled Prevalence and Distribution of *Salmonella* Serotypes

In this study, A total of 145 serotypes out of 919 isolates were identified in humans, food, animals, and the environment (Appendix A). Table 3 shows the pooled prevalence and distribution of *Salmonella* serotypes. The prevalence of *S.* Derby was 23.92% (95% CI: 20.76–27.39), followed by *S.* Croft II at 18.75% (95% CI: 6.17–44.75), *S.* Jodhpur II at 18.75% (95% CI: 6.17–44.75), *S.* Tennessee at 16.35% (95% CI: 10.42–24.73), *S.* Muenster at 10.54% (95% CI: 7.99–13.80), *S.* Typhimurium at 10.42% (95% CI: 7.78–13.83), *S.* Tilene at 10.03% (95% CI: 4.94–19.28), *S.* Jodhpur at 9.77% (95% CI: 3.71–23.34), *S. Chester* at 9.07% (95% CI: 7.01–11.65), and *S.* Hato at 8.31% (95% CI: 6.38–10.75 The prevalence of the remaining *Salmonella* species was less than 8%. The prevalence of *Salmonella* varies considerably between different sources (Table 3). In humans, *S.* Typhimurium was observed at 21.27%, followed by *S.* Poona (15.83%). *S.* Derby was highly prevalent in animals (14.62%), followed by *S.* Muenster (11.26%). Considering environmental sources, *S.* Croft II, *S.* Jodhpur II, and *S.* Kentucky accounted for 18.75%, followed by *S.* Poona (16.13%). S. Derby was highly prevalent in food (30.45%), followed by *S.* Tennessee (27.74%). This study investigated the antimicrobial resistance of different *Salmonella* serotypes (Table 4). Serotype Typhimurium is the only multidrug resistance (MDR) strain found in humans, animals, and food. Moreover, MDR was revealed for the serotypes Brancaster, Enteritidis, Kentucky, Chester, and Derby in food; Hato and Urbana in animals; and Virchow, Poona, Duisburg, and Ouakam in humans.

### 2.5. Antibiotic Resistance Profile of *Salmonella* spp. Isolates

The pooled prevalence of antibiotic resistance for *Salmonella* included in this meta-analysis (Table 5) was as follows: erythromycin had a high prevalence of (98.3%), amoxicillin (68.69%), cefixime (56.24%), cephalothin (54.65%), amoxicillin and clavulanic acid (42.06%), cefepime (38.99%), tetracycline (36.50%), tobramycin (33.68%), cefotaxime (29.44%), colistin sulfate (28.49%), piperacillin (27.49%), and ampicillin (20.84%). The remaining data are shown in Table 5. Chloramphenicol, nalidixic acid, and tetracycline were the most commonly tested antibiotics, with more than 10 studies each. Overall, the MDR recorded for *Salmonella* spp. was 29% [95% CI: 15–50] (Figure 4).

### 2.6. Publication Bias

The Begg and Mazumdar rank correlation test demonstrated no significant publishing bias for all parameters (*p* value > 0.05 (Table 2 and Figure 5).

## 3. Discussion

*Salmonella* is globally one of the leading causes of human death among infectious diarrheal diseases, and antimicrobial-resistant strains of *Salmonella* are a worldwide health concern [49,50,51]. Understanding the epidemiological status of *Salmonella* is thus crucial for controlling this foodborne pathogen [52]. In Burkina Faso, primary research on the prevalence of *Salmonella* in humans, animals, food, and the environment, as well as its antimicrobial resistance status, have been reported. This meta-analysis focused on comprehensive and robust assessments.

This study examined 28 published articles from Burkina Faso, and the majority of the studies included were published between 2008 and 2022. This study revealed that the highest pooled prevalence of *Salmonella* was 23% (95% CI: 21.8–24.4) in Ouagadougou, followed by area 2 [18.4% (95% CI: 14.6–22.8)] and area 1 [6.4% (95% CI: 4.9–8.2)]. The number of studies included might be a possible reason for the region wise pooled prevalence variation. For instance, only a single study each was found in area 3, Nanoro and Nouna, which may not reflect the true pooled prevalence.

The pooled prevalence in the present systemic review and meta-analysis was 4.6% in humans, 20.1% in animals, 16.8% in food, and 15.6% in the environment. One possible reason for the higher prevalence of *Salmonella* in animals and food may be associated with unhygienic husbandry practices [53,54]. Moreover, meat handling practices in slaughterhouses and butcheries are generally unhygienic. The survival of *Salmonella* in both soil and water may be a potential cause of its high prevalence in the environment [55].

In the present study, the pooled prevalence of *Salmonella* among human samples was similar to that reported in studies in West Africa [56] and Ethiopia [57], which were 5.21% (95% CI: 3.37–7.06) and 4.8% (95% CI: 3.9, 5.9), respectively.

However, higher pooled prevalence reports were reported in Morocco (17.9%, 95% CI: 5.7–34.8%), Tunisia (10.2%, 95% CI: 4.3–18.0%), and Sudan (9.2%, 95% CI: 6.5–12.2%). [58]. This variation might be due to the type of population studied, the study period, the isolation method, and seasonal and geographical variations, but it may also increase community awareness of personal and environmental hygiene [59,60].

According to the subgroup analysis, the pooled prevalence of *Salmonella* in animals was 20.1%, which is lower than that reported in Chikwawa and Malawi in pigs (24.6%) [61], but higher than that observed in Nigeria (8.3%) [62], Australia (4%), and Japan (0.5%) [63,64]. The levels of *Salmonella* contamination in animals can vary depending on the country, the nature of the production system and the specific control measures in place [39]. In Burkina Faso, the majority of these animals are still raised under nonmodern conditions. Animals are grazed in an environment with feed of poor microbiological quality. This is more common in poultry farming. These practices undoubtedly lead to *Salmonella* contamination in animals [65].

The pooled prevalence estimate of *Salmonella* in food was similar to, but slightly greater than, that reported for Mali (12.80%) in poultry carcasses [66]. Other studies have reported prevalence rates of 17.7% in Pakistan and 14.4% in Iraq [67]. These results point to food contamination due to noncompliance with good hygiene practices [68]. *Salmonella*, which is usually present in the intestinal tract of animals, can contaminate food during processing procedures and can survive after a cooking process where the cooking time is too short or the temperature is not high enough [69,70]. A study conducted in Burkina Faso showed that 31 of the raw ingredients used to prepare sandwiches were contaminated with *Salmonella* from the water used to wash the lettuce [38]. This contamination can be of environmental, animal, or human origin during the cultivation, harvesting, or handling of plants prior to consumption [38,71].

The environmental samples identified in our meta-analysis consisted mainly of water (well water, canals, taps, fountains, and reservoirs). The reported clustered prevalence of *Salmonella* corroborates that which was reported in Senegal in irrigation water, which was 17% [72]. Increasing evidence indicates that irrigation water is a source (or a vehicle) for the transmission of *Salmonella* [73,74,75,76]. This result is worrying, given that in Ouagadougou, all kinds of wastewater (urban or industrial wastewater) are used in an unplanned and uncontrolled way in urban and peri-urban agricultural activities, including the irrigation of vegetables consumed raw (lettuce in particular) [32]. Moreover, the lack of mineral matter in the soil and the impossibility for some producers to obtain chemical fertilizers have led them to use animal manure as fertilizer, which is more economical and beneficial for the environment. However, several studies have shown that animal excrement is a major vector for bacterial contamination of garden products [77].

The measurement of trends in serovars over time can provide information about emerging serotypes and about the efficacy of prevention [78].

In this study, the majority of strains were isolated from animals (49.8%, 458/919) and food (38.41%, 353/919). This finding is in agreement with a previous report in South Africa [79]. In the present study, NTS from both humans, the environment, food, and animals dominated by 7.69% in *S.* Kentucky, followed by 7.57% in *S.* Cubana, 6.67% in *S.* Poona, and 4.34% in *S.* Senftenberg. *Salmonella* has a broad host range, and hence is considered a universal pathogen. Each serovar has a different ability to adapt to the host environment and cause virulency. Some *Salmonella* serovars are restricted to one host, whereas some have a broad host spectrum [80]. For instance, *S.* derby was isolated from animals and food in 23.92% of the patients. *S.* Croft II and *S.* Jodhpur II were observed only in the environmental samples. In this review, we found that *S.* Typhi was highly prevalent in humans and was also reported in Ethiopia [81], China [82], the Middle East, and North Africa [58]. According to reports from sub-Saharan African nations, *S.* Typhimurium is one of the invasive forms of NTS, especially among susceptible people, such as those with HIV, malnourished children, and malaria [10,83]. Furthermore, MDR strains mainly existed in *S.* Typhimurium. This serovar was also detected in both animal and food samples, even though it was not highly prevalent. Notably, no environmental cases of *S.* Typhi were found.

The main serotypes found in both animal and food samples in this study included *S.* Derby, *S.* Hato, and *S.* Chester. Poona was the most common serotype detected in both human and environmental samples. *Salmonella*, regardless of serovar, can persist in dry environments as well as in water for many weeks to months. Moreover, environmental factors may influence the survival of serovars and could contribute to within- and between-host species differences [84]. Therefore, the ability of a pathogen to spread disease in populations is influenced by host adaptation in many ways [52,85].

Our meta-analysis revealed high NTS resistance against erythromycin (98.3%), amoxicillin (68.69%), cefixime (56.24%), cephalothin (54.65%), and amoxicillin and clavulanic acid (42.06%) for isolates from all source-specific samples. These levels of resistance may be due to the indiscriminate application of antibiotics in human and animal health and food production, followed by the leaching of antibiotics into the environment [86,87]. High resistance of *Salmonella* to erythromycin was observed by Ramatla et al. in a systematic review in South Africa [88]. This is probably due to its veterinary use as a feed supplement and/or treatment, particularly in the poultry sector [89,90]. Amoxicillin is one of the most commonly used antibiotics worldwide for treating salmonellosis [79].

High resistance to this antibiotic was found in a study carried out in the Middle East, where the rate of resistance to amoxicillin was 100% [91].

However, with the emergence of resistant isolates, traditional antibiotics have been replaced by cephalosporins [92]. In the present study, resistance to cefixime and cephalothin was 56.24% and 54.65%, respectively. A meta-analysis carried out in Saudi Arabia reported a high prevalence of 90.9% for cephalothin [93] and 9% for cefixime in Iran [94]. This difference could be explained by the level of antibiotic use, which varies from one country to another.

Ten studies reported multidrug resistance to antimicrobials in the present study. Our results (29%) are in line with those of a previous study [88] in which multidrug resistance (MDR) (28.5%) was detected. It has been documented that infections caused by multidrug-resistant strains are more severe than those caused by susceptible strains [95]. Thus, MDR strains present in food (Typhimurium, Brancaster, Enteritidis, Kentucky, Chester, and Derby) and animals (Typhimurium, Hato, Urbana) could directly threaten human health, as they can cause diseases that are difficult to cure. In addition, integrons, which are mobile genetic elements often associated with multiresistant *Salmonella* phenotypes, are thought to play a key role in the spread of antimicrobial resistance genes among Gram-negative bacteria [96]. The present study shows the urgent need to control the use of antibiotics in veterinary and human medicine to limit the spread of multidrug-resistant *Salmonella* strains.

This study has the following limitations. The number of studies from some localities was very high compared with that from others, which may have influenced the overall estimate. In addition, few studies on this topic in the environment are available. Only one study was included, which limited comparisons of the prevalence and resistance profiles. Some studies reported the total number of isolates without showing the number of individual serotypes. Resistance genotypes were not included in this meta-analysis due to the small number of studies carried out.

## 4. Materials and Methods

### 4.1. Study Design

The current systematic review and meta-analysis were performed according to the Preferred Reporting Items for Systematic Reviews and Meta-Analyses (PRISMA) guidelines for selection criteria, literature search, statistical analysis, and data extraction [97].

### 4.2. Search Strategy

The literature search was performed on four databases, namely, PubMed, Google Scholar, Science Direct, and African Journals Online, and manually by obtaining hard copies of locally published articles directly from authors and local libraries.

The following keywords were used to search for the articles: antibiotic resistance AND antibiotic AND drug resistance AND bacterial resistance AND multidrug resistance AND *Salmonella* species AND human OR animal AND environment OR food AND nontyphoid *Salmonella* AND Burkina Faso. We conducted our last search on November 2022.

### 4.3. Inclusion and Exclusion Criteria

All of the eligible studies included met the following inclusion criteria in the review: articles published primarily on the prevalence of nontyphoidal *Salmonella* spp. in the environment, foods, animals, and humans in Burkina Faso; the type of samples and method of diagnostics used (approved for the detection and antibiogram of *Salmonella* spp.); the exact number of samples as well as the number of positives tested; and articles reported in English only on serotypes and antimicrobial resistance published between 2008 and August 2022. Studies were excluded if they were not performed in Burkina Faso. In addition, reviews with smaller sample sizes (≤50) and articles not reported in English were excluded.

### 4.4. Study Selection

Records identified from various sources with the search strategies were exported to Endnote reference manager software version 8. Duplicate records were identified, recorded, and removed. The titles and abstracts of journal articles were examined and downloaded. Two authors independently screened the title and abstracts with the predefined inclusion criteria. The full versions of potentially relevant articles were obtained to evaluate their eligibility for final inclusion. During the screening process, an additional author was on standby to resolve any discrepancies that may arise with study selection.

### 4.5. Data Extraction and Data Collection

Data including names of authors, publication year, location, total sample size, and standard methods to detect the antibiotic resistant, number of positive samples were collected from each publication independently. Then, they were entered into a spreadsheet (Microsoft Excel^®^), tables, and a word document template. Only *Salmonella* species/isolates/serotypes specific journal articles were included in the meta-analysis. The studies with insufficient data were excluded. In addition, review articles and studies with an abstract, but without a full text, were excluded. The papers that included the drug susceptibility test, and number of multidrug resistance strains were considered and included in this study. 

### 4.6. Data Analysis and Assessment of Risk of Bias

For each study, the prevalence, effect size, and 95% CIs were calculated as point estimates. A comprehensive meta-analysis was used for all of the statistical analyses, version 3.0 (CMA) program (https://www.meta-analysis.com/, accessed on 19 November 2022). The software was used to generate the pooled estimates, Cochran’s Q, *p* values, and forest plots. I^2^ (level of inconsistency) was used to assess the heterogeneity of the studies (Cochran’s Q). I^2^ values of 25–50%, 51–75%, and >75% were considered to represent low, medium, and high heterogeneity, respectively. Low I^2^ values suggest that variability between estimates is consistent with random variation [13]. Interstudy heterogeneity was considered significant if the *p* value of Cochran’s Q test was less than 0.05. The Begg and Mazumdar test was used to investigate the possibility of propagation bias. Funnel plots were used to assess publication bias [98].

## 5. Conclusions

This systematic review and meta-analysis report the trends and distributions of *Salmonella* serotypes and antibiotic resistance in humans, animals, and food in Burkina Faso. It also highlights the lack of published scientific data on antibiotic resistance of *Salmonella* strains in the environment. 

The pooled prevalence of NTS was greatest for animals and food. The predominant NTS serotypes were *S.* Cubana, *S.* Typhimurium, *S.* Derby, *S.* Hato, *S.* Chester, and *S.* Poona. This predominance of emerging isolates is a real public health problem. NTS isolates also were highly resistant to erythromycin, amoxicillin, cefixime, and cephalothin, with a fairly high rate of multidrug resistance. In view of this situation, a consolidated “One Health” approach to the human, animal, food, and environmental sectors is needed to establish a safety net against salmonellosis in Burkina Faso.

## Figures and Tables

**Figure 1 antibiotics-13-00556-f001:**
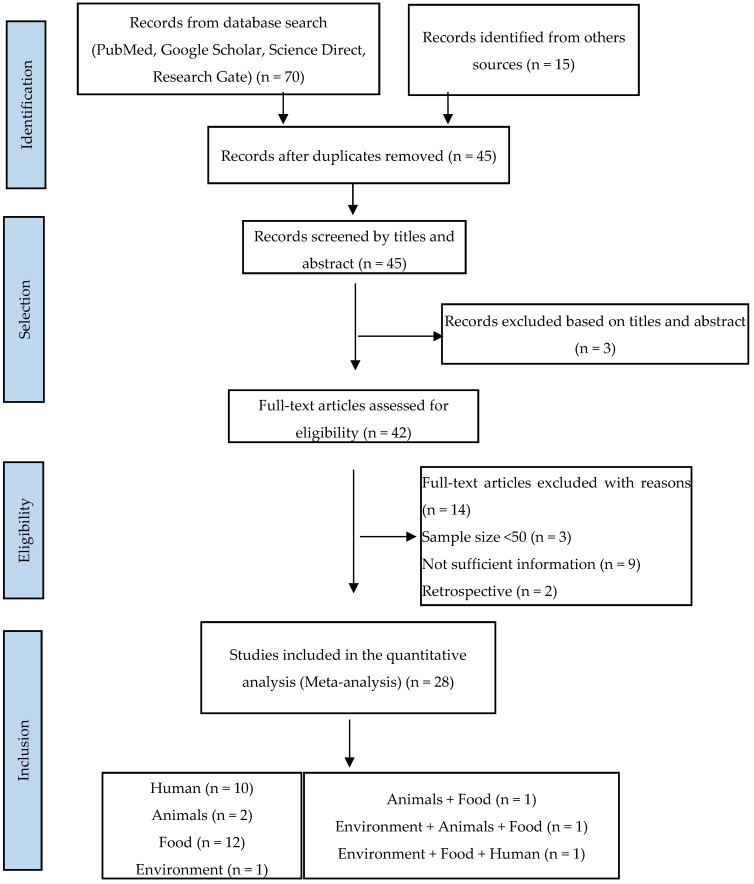
PRISMA flowchart illustrating the process of identifying, screening, and selecting the eligible articles used in this study.

**Figure 2 antibiotics-13-00556-f002:**
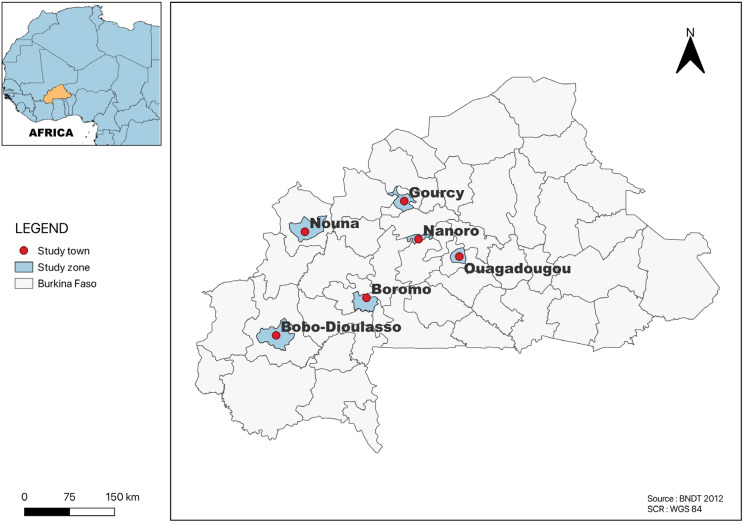
Map of Burkina Faso showing the study area.

**Figure 3 antibiotics-13-00556-f003:**
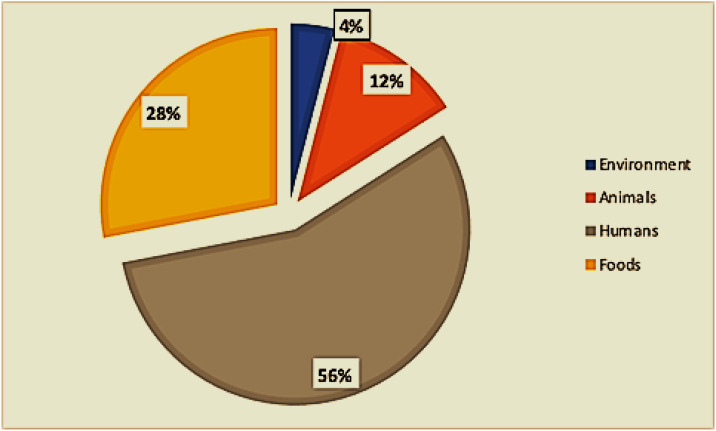
Distribution of sample types.

**Figure 4 antibiotics-13-00556-f004:**
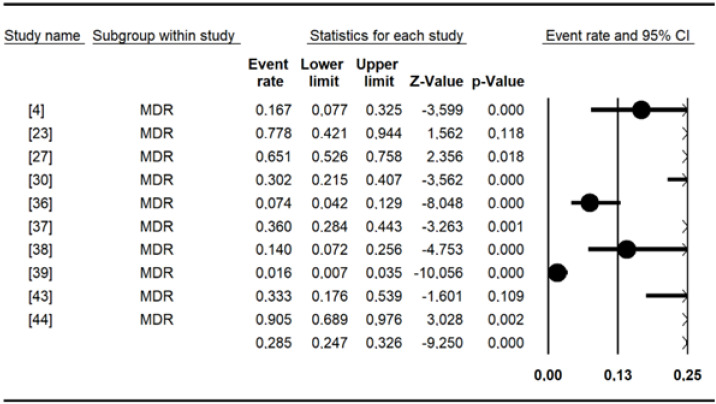
Forest plot showing the pooled estimates of multidrug resistance in studies conducted in Burkina Faso.

**Figure 5 antibiotics-13-00556-f005:**
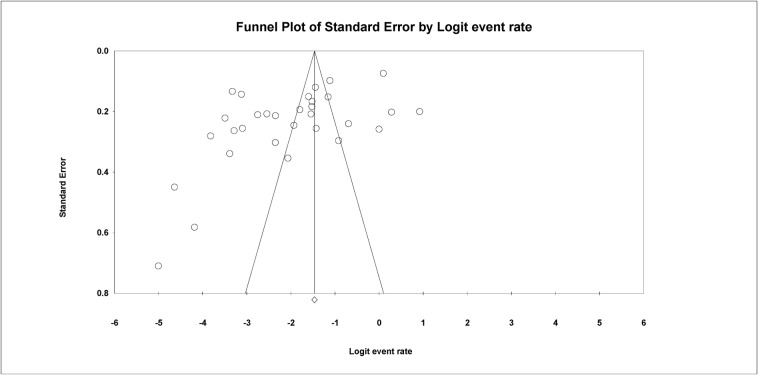
Funnel plot with 95% confidence limits of the pooled prevalence of the studies conducted in Burkina.

**Table 1 antibiotics-13-00556-t001:** Descriptive characteristics of the included studies.

Author, Year	Study Period	Locality	Samples Type	Lab Test Used	Antibiogram	Sample Size	Diagnosis Cases
[4]	2017–2018	Ouagadougou	Food	Culture	DDM	201	36
[27]	2019	Bobo-Dioulasso, Ouagadougou	Food	Culture	DDM	245	44
[28]	2008–2009	Ouagadougou	Food	Culture	DDM	150	19
[26]	2009–2015	Ouagadougou, Gourcy, Boromo	Food	Culture	DDM	78	26
[29]	2010	Ouagadougou	Food	Culture	DDM	100	57
[30]	2011–2012	Ouagadoudou	Food	Culture	DDM	450	86
[31]	2011–2012	Ouagadougou	Food	Culture	DDM	300	2
[32]	2021	Ouagadougou	Food	Culture	DDM	80	9
[33]	2017	Ouagadougou	Food	Culture	Nd	123	88
[34]	2011–2012	Ouagadougou	Food	Culture	DDM	200	3
[35]	2013–2014	Ouagadougou	Food	Culture	Nd	60	30
[15]	2020	Ouagadougou	Food	Culture	Nd	159	28
[36]	2018–2020	Ouagadougou	Food	Culture	DDM	1052	148
[37]	2016	Ouagadougou	Animals	Culture	DDM	563	139
[38]	2008–2009	Ouagadougou	Animals	Culture	DDM	238	57
[27]	2019	Bobo-Dioulasso, Ouagadougou	Animals	Culture	DDM	98	19
[39]	2010	Ouagadougou	Animals	Culture	DDM	729	382
[40]	2010	Ouagadougou	Animals	Culture	DDM	370	16
[38]	2008–2009	Ouagadougou	Environment	Culture	DDM	218	31
[26]	2009–2015	Ouagadougou, Gourcy, Boromo	Environment	Culture	DDM	56	16
[41]	2008–2009	Ouagadougou	Environment	Culture	DDM	138	12
[42]	2009–2010	Ouagadougou	Humans	Culture	DDM	343	25
[23]	2009–2010	Gourcy, Boromo	Humans	Culture	Nd	275	9
[26]	2009–2015	Ouagadougou, Gourcy, Boromo	Humans	Culture	DDM	416	15
[43]	2009–2010	Gourcy, Boromo	Humans	Culture	DDM	400	24
[44]	2012–2013	Ouagadougou	Humans	Culture	DDM	711	21
[45]	2015	Nanoro	Humans	Culture	DDM	1207	51
[46]	2013	Ouagadougou	Humans	Culture	Nd	200	3
[47]	2015	Ouagadougou	Humans	Culture	Nd	1674	58
[25]	2013–2017	Province de Kossi	Humans	Culture + PCR	Nd	605	13
[48]	2010	Gourcy, Boromo	Humans	Culture	Nd	275	24
[24]	2013–2015	Ouagadougou	Humans	Culture	Nd	315	53

Legend: DDM = Kirby Bauer Disc Diffusion Method, Nd = Not performed, PCR = Polymerase Chain Reaction.

**Table 2 antibiotics-13-00556-t002:** Pooled prevalence of *Salmonella* spp. from the environment, animals, food, and humans and sampling sites.

Risk Factors	Pooled Estimates		Measure of Heterogeneity			Publication Bias Begg and Mazumdar Rank *p* Value
	Number of Studies	Pooled Prevalence (95% CI)	Q	I^2^	*p* Value	
Sources						
Humans	11	5.4 (4.8–6)	124.138	91.94	0.00	0.64
Animals	04	20.1 (6.6–47.4)	196.905	98.48	0.00	0.30
Food	14	16.6 (9.9–26.4)	333.514	96.10	0.00	0.10
Environment	03	15.6 (8.2–27.5)	11.907	83.20	0.003	1.00
Locality						
Ouagadougou	24	23.1 (21.9–24.4)	1152.833	98.00	0.00	0.06
Area 1	03	6.4 (4.9–8.2)	6.877	70.92	0.032	1.00
Area 2	02	18.4 (14.6–22.8)	0.095	0.00	0.76	-
Area 3	01	10.4 (8.1–13.2)	-	-	-	-
Nanoro	01	4.2 (3.2–5.5)	-	-	-	-
Nouna	01	2.1 (1.3–3.7)	-	-	-	-

Legend: Area 1: Boromo and Gourcy; Area 2: Bobo-Dioulasso and Ouagadougou; Area 3: Ouagadougou, Boromo and Gourcy.

**Table 3 antibiotics-13-00556-t003:** Pooled prevalence and distribution of *Salmonella* serotypes.

*Salmonella* Serotypes Prevalence					Source-Specific Prevalence % (95% CI)			
Serotypes	No. of Study	Prevalence % (95% CI)	Q I^2^	I^2^	Human	Animal	Environment	Food
*S.* Derby	9	23.92 (20.76–27.39)	71.76	88.85	-	14.62 (11.54–18.36)	-	30.45 (19.80–43.70)
*S.* Croft II	1	18.75 (6.17–44.75)	0.00	0.00	-	-	18.75 (6.17–44.75)	-
*S.* Jodhpur II	1	18.75 (6.17–44.75)	0.00	0.00	-	-	18.75 (6.17–44.75)	-
*S.* Tennessee	3	16.35 (10.42–24.73)	46.21	95.67	-	0.79 (0.25–2.41)		27.74 (6.89–66.59)
*S.* Muenster	4	10.54 (7.99–13.80)	3.35	10.32	5.26 (1.32–18.76)	11.26 (8.45–14.84)	--	2.78 (0.39–17.26)
*S.* Typhimurium	5	10.42 (7.78–13.83)	39.29	89.82	21.27 (13.45–31.96)	2.09 (1.05–4.13)	-	12.03 (7.80–18.08)
*S.* Tilene	4	10.03 (4.94–19.28)	11.50	73.92	-	1.75 (0.25–11.43)	3.23 (0.45–19.64)	10.50 (0.99–57.81)
*S.* Jodhpur	2	9.77 (3.71–23.34)	0.26	0.00	-	-	12.50 (3.14–38.60)	7.69 (1.93–26.07)
*S.* Chester	4	9.07 (7.01–11.65)	2.39	0.00	-	8.38 (5.99–11.61)	3.23 (0.45–19.64)	10.78 (7.13–15.97)
*S.* Hato	7	8.31 (6.38–10.75)	6.78	11.50	-	7.63 (5.38–10.71)		9.12 (5.75–14.15)
*S.* Kentucky	7	7.69 (4.56–12.67)	14.27	57.97	6.67 (0.93–35.20)	1.75 (0.25–11.43)	18.75 (6.17–44.75)	4.53 (1.23–15.29)
*S.* Cubana	5	7.57 (3.81–14.48)	3.89	0.00	8.93 (2.89–24.47)	1.75 (0.25–11.43)	12.50 (3.14–38.60)	7.69 (1.93–26.07)
*S.* Virchow	4	7.03 (3.50–13.63)	14.40	79.17	13.33 (5.04–30.84)	0.26 (0.04–1.83)	9.68 (3.15–26.06)	-
*S.* Drac	2	6.86 (4.84–9.64)	0.25	0.00	-	6.86 (4.84–9.64)	-	-
*S.* Hvittingsfoss	2	6.74 (1.68–23.41)	0.51	0.00	6.74 (1.68–23.41)	-	-	-
*S.* Poona	7	6.67 (4.24–10.33)	26.11	77.02	15.83 (7.69–29.81)	1.31 (0.55–3.11)	16.13 (6.88–33.37)	3.27 (0.82–12.15)
*S.* Colindale	2	5.69 (2.34–13.22)	12.61	92.07	-	0.26 (0.04–1.83)	12.90 (4.93–29.74)	-
*S.* Essen	3	4.60 (1.90–10.68)	5.63	64.48	-	-	12.50 (3.14–38.60)	2.07 (0.34–11.61)
*S.* Bredeney	5	4.55 (2.87–7.12)	11.71	65.85	-	4.02 (2.33–6.83)	3.23 (0.45–19.64)	7.42 (2.81–18.17)
*S.* Duisburg	3	4.26 (1.37–12.48)	1.77	0.00	6.74 (1.68–23.41)	1.75 (0.25–11.43)	-	-
*S.* Nima	3	3.36 (1.74–6.41)	14.46	86.17	-	1.05 (0.39–2.76)	-	6.03 (0.73–35.88)
*S.* Brancaster	6	3.34 (2.08–5.32)	9.80	48.96	-	2.19 (0.98–4.83)	-	4.20 (2.34–7.42)
*S.* Agona	5	2.28 (1.19–4.34)	6.44	37.89	-	2.01 (0.90–4.42)	-	2.94 (0.95–8.73)
*S.* Ouakam	5	1.77 (0.73–4.21)	8.67	53.85	6.74 (1.68–23.41)	0.26 (0.04–1.83)	-	1.28 (0.32–4.96)
*S.* Eastbourne	4	1.57 (0.87–2.82)	1.11	0.00	-	1.56 (0.84–2.87)	-	1.75 (0.25–11.43)
*S.* Fresno	3	1.43 (0.64–3.15)	1.59	0.00	4.17 (0.58–24.35)	1.16 (0.48–2.76)	-	-
*S.* Korlebu	2	0.78 (0.25–2.40)	0.98	0.00	-	0.78 (0.25–2.40)	-	-

**Table 4 antibiotics-13-00556-t004:** Multidrug-resistant *Salmonella* serotypes from different sources *.

*Salmonella* Serotypes	Origine		
	Animals	Food	Humans
Brancaster			
Enteritidis			
Kentucky			
Typhimurium			
Hato			
Urbana			
Chester			
Derby			
Groupe Z			
Groupe B			
Virchow			
Poona			
Duisburg			
Ouakam			
*Salmonella* spp.			

Legend: The black box indicates multidrug-resistant *Salmonella* serotypes. * Data reported from the 28 references.

**Table 5 antibiotics-13-00556-t005:** Percentage of pooled resistance rates of antimicrobials to *Salmonella* isolates from humans, animals, food, and the environment.

Antibiotics	No. of Study	Prevalence % (95%)	Q	I^2^	Source-Specific Prevalence % (95% CI)			
					Animals	Food	Humans	Environment
Nalidixic acid	11	12.62 (9.88–16.00)	44.69	77.62	0.71 (0.21–2.44)	13.75 (10.36–18.02)	17.08 (10.15–27.32)	-
Amoxicillin	2	96.01 (87.16–98.84)	0.03	0	-	-	96.01 (87.16–98.84)	-
Amoxicillin and clavulanic acid	9	46.41 (41.08–51.82)	128.95	93.8	52.63 (31.11–73.22)	41.67 (35.97–47.60)	72.30 (58.81–82.68)	-
Ampicillin	6	15.26 (11.77–19.54)	59.2	91.55	1.83 (0.88–3.79)	15.17 (10.75–20.99)	41.92 (28.73–56.37)	-
Aztreonam	8	21.78 (17.35–26.97)	49.74	85.93	5.26 (0.74–29.39)	14.72 (10.84–19.68)	43.91 (32.38–56.13)	-
Cephalothin	1	54.65 (44.08–64.82)	0	0	-	54.65 (44.08–64.82)	-	-
Cefamandole	1	13.95 (8.10–22.98)	0	0	-	13.95 (8.10–22.98)	-	-
Cefepime	3	56.30 (43.86–68.00)	8.91	77.55	-	-	56.30 (43.86–68.00)	-
Cefixime	2	49.82 (37.52–62.14)	1.13	11.48	-	-	49.82 (37.52–62.14)	-
Cefotaxime	3	45.75 (35.20–56.70)	0.99	0	36.84 (18.68–59.70)	-	48.38 (36.27–60.69)	-
Ceftriaxone	7	21.26 (15.58–28.31)	70.85	91.53	5.26 (0.74–29.39)	3.80 (2.05–6.91)	48.57 (36.64–60.67)	-
Cephalexin	3	10.91 (7.29–16.03)	11.16	82.08	-	10.91 (7.29–16.03)	-	-
Chloramphenicol	11	15.56 (12.15–19.71)	134.85	92.58	1.63 (0.76–3.47)	3.23 (1.92–5.38)	43.78 (34.77–53.22)	-
Ciprofloxacin	8	11.85 (8.02–17.16)	17.21	59.32	5.26 (0.74–29.39)	9.43 (5.24–16.37)	15.77 (9.14–25.84)	-
Colistin sulfate	5	26.68 (21.40–32.72)	21.17	81.1	15.79 (5.18–39.15)	18.58 (13.59–24.88)	48.38 (36.27–60.69)	-
Erythromycin	2	98.34 (89.10–99.77)	0.17	0	97.50 (70.19–99.85)	98.89 (84.57–99.93)	-	-
Gentamicin	6	21.06 (16.93–25.88)	10.92	54.2	-	20.46 (15.88–25.95)	23.02 (14.73–34.10)	-
Imipenem	2	25.80 (14.68–41.27)	12.19	91.8	-	-	25.80 (14.68–41.27)	-
Kanamycin	1	5.56 (1.39–19.67)	0	0	-	5.56 (1.39–19.67)	-	-
Mecillinam	1	0.52 (0.13–2.07)	0	0	0.52 (0.13–2.07)	-	-	-
Netilmicin	1	1.89 (0.27–12.21)	0	0	-	-	1.89 (0.27–12.21)	-
Norfloxacin	2	20.89 (10.04–38.45)	3.81	73.78	-	7.69 (1.93–26.07)	33.33 (14.60–59.40)	-
Piperacillin	2	30.79 (20.56–43.33)	0.35	0	-	-	30.79 (20.56–43.33)	-
Piperacillin and Tazobactam	1	1.89 (0.27–12.21)	0	0	-	-	1.89 (0.27–12.21)	-
Spectinomycin	1	8.33 (2.71–22.86)	0	0	-	8.33 (2.71–22.86)	-	-
Streptomycin	8	11.64 (9.39–14.36)	14.46	51.58	9.51 (6.98–2.82)	12.39 (8.74–17.27)	33.33 (17.63–53.88)	6.45 (1.62–22.42)
Sulfonamides	4	10.01 (6.68–14.75)	36.35	91.75	1.83 (0.88–3.79)	16.37 (8.74–28.56)	33.33 (17.63–53.88)	-
Tetracycline	14	37.23 (32.53–42.18)	248.35	94.77	7.35 (4.35–2.14)	38.70 (32.98–44.74)	70.75 (59.36–80.02)	-
Ticarcillin	3	22.46 (12.72–36.53)	14.55	86.25	42.11 (22.63–64.39)	7.93 (2.94–19.67)	-	-
Tobramycin	2	33.87 (23.24–46.43)	0	0	-	-	33.87 (23.24–46.43)	-
Trimethoprim	2	7.56 (4.36–12.80)	33.25	96.99	1.57 (0.71–3.45)	-	33.33 (17.63–53.88)	-
Trimethoprim and Sulfamethoxazole	9	16.24 (12.36–21.05)	56.45	85.83	10.53 (2.65–33.74)	5.54 (3.50–8.65)	36.33 (26.77–47.12)	
MDR	10	29 (15–50)		94.34	-	-	-	-

## Data Availability

The data presented in this study are available upon request from the corresponding author.

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
