# Peer review of "Characteristics of Nontyphoid Salmonella Isolated from Human, Environmental, Animal, and Food Samples in Burkina Faso: A Systematic Review and Meta-Analysis"

_antibiotics, 2024, doi:10.3390/antibiotics13060556_

Round 1

Reviewer 1 Report

Comments and Suggestions for Authors

The information in the manu is needed for setting up a safety net against salmonellosis in Burkina Faso.

Comments on the Quality of English Language

The language is fine, and easy reading.

Author Response

Reviewer #1: The information in the manu is needed for setting up a safety net against salmonellosis in Burkina Faso.

Author: Dear Reviewer, thank you for your comment.

Reviewer 2 Report

Comments and Suggestions for Authors

Comments for the authors

Table 1: The table needs to be reset because the page setup numbers are overwritten.

Table 4: the same as in table 1, furthermore in the caption you wrote “The black box indicates multidrug-resistant Salmonella serotypes, *Data reported from the 29 references (..)” why 29? In the paper you wrote that 28 paper have been included, is quite confusing, please revise.

In line 101 it is stated: “Of the total, 28 publications (33%) considered suitable for inclusion in this review”

Line 361: “this study examined 29 published articles from Burkina Faso and the majority of the studies included were conducted between 2008 to 2021”.

Please explain or correct, 29 or 28 ?

As mentioned by the authors, this study has some limitations:  the resistance genotypes were not included, and this would have been interesting, the studies in the environmental field are limited.

Systematic reviews and meta- analysis, are current and  very useful to estimate and have an overview of the topic in wich you want to work or you are working, but here the number of retrieve it is very poor, due also to the country in wich they are concentrated.

I would reccomend to take into consideration the lack of data on enviromental field in the discussion, but also in further research.

Comments on the Quality of English Language

The english is clear and well written, but as always, I ask the authors to have their paper read by a native speaker. 

Author Response

 Reviewer #2: Table 1: The table needs to be reset because the page setup numbers are overwritten.

Author: Dear Reviewer, thank you for your comment. It is done

Reviewer #2: Table 4: the same as in table 1, furthermore in the caption you wrote “The black box indicates multidrug-resistant Salmonella serotypes, *Data reported from the 29 references (..)” why 29? In the paper you wrote that 28 paper have been included, is quite confusing, please revise.

Author: It is done. The Data reported from the 29 references was an error on our part, and has been corrected.

Reviewer #2: In line 101 it is stated: “Of the total, 28 publications (33%) considered suitable for inclusion in this review”

Line 361: “this study examined 29 published articles from Burkina Faso and the majority of the studies included were conducted between 2008 to 2021”.

Please explain or correct, 29 or 28 ?

Author: It was an error on our part. It has been corrected. It is done

Reviewer #2: As mentioned by the authors, this study has some limitations:  the resistance genotypes were not included, and this would have been interesting, the studies in the environmental field are limited.

Systematic reviews and meta- analysis, are current and  very useful to estimate and have an overview of the topic in wich you want to work or you are working, but here the number of retrieve it is very poor, due also to the country in wich they are concentrated.

I would reccomend to take into consideration the lack of data on enviromental field in the discussion, but also in further research.

Author: Thank you to the Reviewer for this suggestion. However, the lack of environmental data has already been taken into account within the limits of the study in the discussion. It has also been taken into account as a perspective for further research in the conclusion.

Reviewer #2: Comments on the Quality of English Language: The english is clear and well written, but as always, I ask the authors to have their paper read by a native speaker. 

Author: Dear Reviewer, the language of this manuscript, which is English, has been revised

Reviewer 3 Report

Comments and Suggestions for Authors

The objective of this review was to present knowledge on the prevalence, serovars and antimicrobial resistance phenotypes of strains of non-typhoidal Salmonella serotypes from Burkina Faso.

The authors must present and explain what gaps in the international literature are filled by this manuscript. This is not made clear at all.

The authors must alsi underline the novelty offered by this manuscript. What are the newly offered advancements by this manuscript in comparison to existing knowledge and previous publications?

The authors also must as control studies similar reports about presence of these bacteria in other neighbouring countries. Also, please explain differences in the literature between papers from Burkina Faso and papers from neighbouring countries. Moreover, the authors must present similar types of studies regarding papers dealing with other pathogens from Burkina Faso.

The above suggestions must be incorporated and that way there will be some type of ‘control’ papers to allow comparisons and best practices in the selection and analysis of the literature references.

In supplementary material, please provide a list of the references found and analysed.

Can the authors please provide a chronological evaluation of the papers in accord with the Salmonella serotypes reported in there?

Finally, the concluding section must be rewritten to be in better concordance with the findings of the study.

Comment about Tables. These are OK. Please note the comment above regarding additional appendices.

Comment about Figures. The figures are not good quality. Hence, they must be improved. Further pictures must be added to improve the visualisation of the results.

General. The manuscript requires extensive improvements. As it is, it cannot be accepted. Please revise by taking into account the above and resubmit an improved version.

Author Response

The objective of this review was to present knowledge on the prevalence, serovars and antimicrobial resistance phenotypes of strains of non-typhoidal Salmonella serotypes from Burkina Faso.

 Reviewer #3: The authors must present and explain what gaps in the international literature are filled by this manuscript. This is not made clear at all.

Author: Thank you for your suggestion Dear Reviewer. As you can see, some

sentences in the manuscript have been reworded.

 Reviewer #3: The authors must also underline the novelty offered by this manuscript. What are the newly offered advancements by this manuscript in comparison to existing knowledge and previous publications?

Author: Thank you for your suggestion Dear Reviewer. As you can see, some

sentences in the manuscript have been reworded.

 Reviewer #3: The authors also must as control studies similar reports about presence of these bacteria in other neighbouring countries. Also, please explain differences in the literature between papers from Burkina Faso and papers from neighbouring countries. Moreover, the authors must present similar types of studies regarding papers dealing with other pathogens from Burkina Faso.

The above suggestions must be incorporated and that way there will be some type of ‘control’ papers to allow comparisons and best practices in the selection and analysis of the literature references.

Author: Thank you for your suggestion. However, we recall that the aim of this systematic review and meta-analysis was to take stock of the current state of knowledge on salmonellosis in Burkina Faso, and to identify gaps in research on these bacteria, which is necessary for the development of prevention and control methods. In addition, the use of similar reports on the presence of these bacteria in other neighbouring countries as control studies is highly relevant to the discussion. However, there are very few systematic studies on this pathogen in the sub-region. Systematic studies carried out in the sub-region have already been used in this review.

 Reviewer #3: In supplementary material, please provide a list of the references found and analyzed.

Author: It is done

 Reviewer #3: Can the authors please provide a chronological evaluation of the papers in accord with the Salmonella serotypes reported in there?

Author: Thank you for your suggestion. Our data does not allow us to do this

 Reviewer #3: Finally, the concluding section must be rewritten to be in better concordance with the findings of the study.

 Author: Thank you for this comment. As you can see, some sentences in the conclusion have been reworded.

 Reviewer #3: Comment about Tables. These are OK. Please note the comment above regarding additional appendices.

Comment about Figures. The figures are not good quality. Hence, they must be improved. Further pictures must be added to improve the visualisation of the results.

 General. The manuscript requires extensive improvements. As it is, it cannot be accepted. Please revise by taking into account the above and resubmit an improved version.

Author: Thank you for this comment, but we only have the images that were submitted for this review

Round 2

Reviewer 3 Report

Comments and Suggestions for Authors

The changes made by the authors have improved the manuscript. I am a bit concerned however about the quality of figures in the manuscript. Can these be replaced with better figures, please?